# Progesterone Receptor Membrane Component 1 (PGRMC1) Modulates Tumour Progression, the Immune Microenvironment and the Response to Therapy in Glioblastoma

**DOI:** 10.3390/cells12202498

**Published:** 2023-10-20

**Authors:** Claudia Alexandra Dumitru, Hannah Schröder, Frederik Till Alexander Schäfer, Jan Friedrich Aust, Nina Kreße, Carl Ludwig Raven Siebert, Klaus-Peter Stein, Aiden Haghikia, Ludwig Wilkens, Christian Mawrin, Ibrahim Erol Sandalcioglu

**Affiliations:** 1Department of Neurosurgery, Otto-von-Guericke University, 39120 Magdeburg, Germanyerol.sandalcioglu@med.ovgu.de (I.E.S.); 2Department of Neurology, Otto-von-Guericke University, 39120 Magdeburg, Germany; 3Department of Pathology, Nordstadt Hospital Hannover, 30167 Hannover, Germany; 4Department of Neuropathology, Otto-von-Guericke University, 39120 Magdeburg, Germany

**Keywords:** glioblastoma, PGRMC1, cancer progression, individualized therapy, neutrophils

## Abstract

Progesterone Receptor Membrane Component 1 (PGRMC1) is a tumour-promoting factor in several types of cancer but its role in brain tumours is poorly characterized thus far. Our study aimed to determine the effect of PGRMC1 on glioblastoma (GBM) pathophysiology using two independent cohorts of IDH wild-type GBM patients and stable knockdown GBM models. We found that high levels of PGRMC1 significantly predicted poor overall survival in both cohorts of GBM patients. PGRMC1 promoted the proliferation, anchorage-independent growth, and invasion of GBM cells. We identified Integrin beta-1 (ITGB1) and TCF 1/7 as potential members of the PGRMC1 pathway in vitro. The levels of ITGB1 and PGRMC1 also correlated in neoplastic tissues from GBM patients. High expression of PGRMC1 rendered GBM cells less susceptible to the standard GBM chemotherapeutic agent temozolomide but more susceptible to the ferroptosis inducer erastin. Finally, PGRMC1 enhanced Interleukin-8 production in GBM cells and promoted the recruitment of neutrophils. The expression of PGRMC1 significantly correlated with the numbers of tumour-infiltrating neutrophils also in tissues from GBM patients. In conclusion, PGRMC1 enhances tumour-related inflammation and promotes the progression of GBM. However, PGRMC1 might be a promising target for novel therapeutic strategies using ferroptosis inducers in this type of cancer.

## 1. Introduction

Glioblastoma (GBM) is the most prevalent malignant brain tumour affecting adults [1]. The vast majority of GBM develop de novo (primary GBM) and are more frequently observed in male individuals [2]. GBM patients have a dismal survival rate with less than 7% surviving beyond five years [1,2]. The current protocol for GBM treatment involves surgically removing as much of the tumour mass as safely possible, followed by a combination of radiation therapy and chemotherapy with the alkylating agent temozolomide (TMZ) [2,3,4]. Many phase II–IV clinical trials targeted receptor tyrosine kinases, in particular EGFR, as well as angiogenesis via the VEGF/VEGFR system (reviewed in [5]). Other clinical studies additionally targeted the stem cell pathways (Wnt, Notch, Hedgehog), the extrinsic and intrinsic apoptosis pathways (CD95, Bcl-2), autophagy or the cell cycle, and DNA repair pathways (CDK 4/6, PARP) [5]. Most of these multimodal therapeutic strategies have met, however, only with very limited success. Thus, there is an urgent need to identify novel factors that control the progression of GBM and could ultimately serve as effective therapeutic targets in this type of cancer. 

Progesterone Receptor Membrane Component 1 (PGRMC1) is a heme-binding protein, which belongs to the membrane-associated progesterone receptor (MAPR) family of cytochrome b5-related proteins [6]. PGRMC1 is involved in a variety of biological processes, including non-genomic P4 responses in the female reproductive tract, axon guidance during embryological formation of the central nerve cord, membrane trafficking, cytochrome P450-mediated steroidogenesis, lipid synthesis and stress-response (reviewed in [7]). Accumulating evidence indicates that PGRMC1 plays important roles in the biology of cancer as well. Specifically, PGRMC1 is overexpressed in tumours compared to healthy tissues and associates with the poor outcome of patients with breast, head and neck, hepatocellular, or renal cancer [8,9,10,11] (reviewed in [6]). Additionally, there is a consensus that PGRMC1 leads to resistance against different chemotherapeutic drugs, including alkylating agents such as cisplatin [12,13,14,15,16]. This effect of PGRMC1 is mainly attributed to its dimerization and subsequent binding to cytochrome P450 [17]. Interestingly, very recent studies found that ferroptosis inducers could efficiently eliminate paclitaxel-resistant head and neck cancer cells, a phenomenon that was a consequence of high PGRMC1 levels [18]. These findings suggest that PGRMC1––otherwise a pro-tumour factor––could be ‘hijacked’ by novel therapeutics to acquire anti-tumour properties.

Recent evidence additionally linked PGRMC1 with a pro-inflammatory immune response in hepatocellular carcinoma [8]. These findings are of particular interest, since the interplay between tumour cells and the immune system is emerging as an important regulator of GBM pathophysiology. Indeed, glioma and GBM tissues are infiltrated by immune cells—the majority of which are myeloid cells, namely macrophages and neutrophils (reviewed in [19]). Importantly, the presence of infiltrating neutrophils significantly associates with increased glioma malignancy and a poor outcome in these patients [19,20,21], which suggests that neutrophils contribute to the progression of glioma and GBM. The exact mechanisms mediating the interactions between neutrophils and GBM cells remain, however, to be elucidated.

There is currently very little known about the role of PGRMC1 in GBM. Our study aimed to characterize: (1) the association between PGRMC1 expression and the clinical outcome of GBM patients, (2) the effect of PGRMC1 on the intrinsic functions of the GBM cells, (3) the effect of PGRMC1 on the response of GBM cells to treatment with TMZ and ferroptosis inducers, and (4) the effect of PGRMC1 on GBM-induced modulation of neutrophil biology.

## 2. Materials and Methods

### 2.1. Study Subjects

We conducted a retrospective analysis on tissues from two independent cohorts of adult patients with newly diagnosed, IDH wild type GBM. The patients in the Hannover cohort received treatment at the Department of Neurosurgery, Nordstadt Hospital Hannover from 2004 to 2014 while in the Magdeburg cohort, the patients were treated at the Department of Neurosurgery, University Hospital Magdeburg between 2005 and 2018. The median age of the patients was 67 years in both cohorts. The studies were conducted in accordance with the Declaration of Helsinki issued in 1975 (revised in 2013) and received approval from the ethics committees of the Medical School Hannover (Study No. 6864, 2015) and Otto-von-Guericke University Magdeburg (Study No. 146, 2019), respectively. The ethics committees granted a waiver for obtaining informed consent. The clinical characteristics of the patients are summarized in Appendix A. 

### 2.2. Tissue Microarrays (TMA): Immunohistochemistry and Scoring

TMAs were constructed and stained as previously described [22,23]. The following primary antibodies were used: 112 ng/mL monoclonal rabbit anti-PGRMC1 (Cell Signaling Technology, Frankfurt am Main, Germany), 3.6 µg/mL polyclonal rabbit anti-Integrin beta-1 (Proteintech Europe, Manchester, UK), and 0.66 µg/mL monoclonal mouse anti-CD66b (BioLegend, San Diego, CA, USA). The stained TMAs were digitally captured with a high-resolution whole slide scanner (Aperio VERSA, Leica Biosystems, Nussloch, Germany). The resulting digital images were examined with the Aperio ImageScope V12.1.0.5029 software (Leica Biosystems). Blinded histological analysis was independently conducted by authors CAD, HS, FTAS, JFA, NK, and CLRS.

PGRMC1 and Integrin beta-1 predominantly displayed a cytoplasmic subcellular localization. The intensity of marker expression was classified as ‘weak’, ‘medium’, or ‘strong’ and received 1, 2, or 3 points, respectively (Appendix A). Considering the variations in the staining pattern among samples, the expression levels were further assessed using the H-Score according to the formula: 

(1 × X) + (2 × Y) + (3 × Z), where X + Y + Z = 100% of the total tumour area. 

### 2.3. Cells and Supernatants

We used the following GBM cell lines: H4 (RRID:CVCL_5575), U343-MG (RRID:CVCL_S471) and U251-MG (RRID:CVCL_0021). These cells were generously provided by Prof. A. Temme (University Hospital Dresden, Dresden, Germany), but are also available commercially. All cell lines have been authenticated using STR profiling within the last three years. The cells were cultured in Dulbecco’s Modified Eagle Medium (DMEM; ThermoFisher Scientific, Waltham, MA, USA) additionally containing 10% fetal calf serum (Pan Biotech, Aidenbach, Germany), and 1% Penicillin-Streptomycin (ThermoFisher Scientific). Mycoplasma testing was performed routinely on the cultured cells and showed no contamination. The GBM cells were transfected with either the OmicsLink^TM^ shRNA clone HSH091094-nH1-a targeting PGRMC1 or with CSHCTR001-nH1 as control (both from GeneCopoeia, Rockville, MD, USA). The transfection was performed in antibiotics-free medium using PANFect A-plus transfection reagent (Pan Biotech) as indicated by the manufacturer. Transfected cells were selected with 1 µg/mL Puromycin (InvivoGen, Toulouse, France) and were subsequently maintained in cell culture medium containing 0.3 µg/mL (H4 cells) or 0.5 µg/mL (U343 and U251 cells) Puromycin. The efficacy of PGRMC1 knockdown (sh-PGRMC1) was evaluated both at protein and mRNA levels compared to control transfection (sh-control) (Appendix A).

To generate conditioned supernatants, GBM cells (10^6^ cells/mL) were cultured for 24 h at 37 °C in DMEM supplemented as above. The supernatants were freed by cell debris by centrifugation, divided into aliquots and stored at −20 °C for later use.

Neutrophils were isolated from the peripheral blood of healthy volunteers as previously described in [20]. Briefly, EDTA-anticoagulated venous blood was diluted 1:1 *v*/*v* with phosphate-buffered saline (PBS) (Carl Roth, Karlsruhe, Germany) and subjected to density gradient centrifugation using Pancoll (Pan Biotech). The neutrophils were further separated from contaminating erythrocytes by a sedimentation step with 1% PVA solution (polyvinyl alcohol, Carl Roth) for 25 min at room temperature followed by a short (30 s) osmotic shock with distilled water. The purity of the isolated neutrophil population routinely exceeded 98%.

### 2.4. SDS-PAGE and Western Blot

GBM cells were subjected to lysis using a buffer containing Triton X-100 along with protease and phosphatase inhibitors (Cell Signaling Technology). The lysates were cleared by cellular debris through centrifugation and then mixed with a loading buffer containing 4% glycerin, 0.8% SDS, 1.6% beta-mercaptoethanol, and 0.04% bromophenol blue (all from Carl Roth). The proteins were electrophoretically separated by SDS-PAGE and then transferred to Immobilon-P (Merck Millipore, Darmstadt, Germany) or Roti^®^-Fluoro (Carl Roth) PVDF membranes using a semidry blotting device (Biometra Fastblot^TM^, Analytik Jena AG, Jena, Germany). The membranes were incubated overnight at 4 °C with the following primary antibodies: anti-PGRMC1, anti-TCF 1/7, anti-beta-Actin (all from Cell Signaling Technology), or anti-ITGB1 (Proteintech, Manchester, UK). Subsequent secondary reactions were carried out for 1 h at room temperature with HRP-, AlexaFluor^®^488-, or AlexaFluor^®^647-coupled antibodies (all from Cell Signaling Technology). All antibodies were diluted in SignalBoost™ Immunoreaction Enhancer (Merck Millipore) according to the manufacturer’s recommendation. The ChemoStar imaging system (Intas Science Imaging, Göttingen, Germany) was used for the detection and acquisition of signals and the intensity of the bands was quantified with the ImageJ 1.48v software.

### 2.5. Gene Expression Analysis

The mRNA from both sh-PGRMC1 and sh-control GBM cells was extracted with the InnuPREP RNA Mini Kit 2.0 (Analytik Jena) according to the manufacturer’s instructions. Subsequent reverse transcription was performed with the LunaScript RT SuperMix Kit (New England Biolabs, Frankfurt am Main, Germany) under the following thermal conditions: 2 min at 25 °C, 20 min at 55 °C, and 1 min at 95 °C. The samples were incubated with the following primers against PGRMC1 and GAPDH in the presence of Luna Universal qPCR Mix (New England Biolabs):

PGRMC1 forward 5′-ACGGCAAGGTGTTCGATGTG-3′

PGRMC1 reverse 5′-GGCAGCAGTGAGGTCAGAAAG-3′

GAPDH forward 5′-AGGGCTGCTTTTAACTCTGGT-3′

GAPDH reverse 5′-CCCCACTTGATTTTGGAGGGA-3′

After an initial denaturation for 1 min at 95 °C, the samples underwent 40 thermal cycles of 15 s at 95 °C and 30 s at 60 °C, respectively. The expression of *PGRMC1* was normalized to *GAPDH* using the ΔΔCq method. The analysis was performed with the Bio-Rad CFX Maestro V4.1.2433.1219 software (Bio-Rad, Feldkirchen, Germany).

### 2.6. MTT Assay

GBM cells were seeded in 96-well plates at a density of 2000 cells/well. At the specified time intervals, fresh medium containing 10% MTT (3-(4,5-dimethylthiazol-2-yl)-2,5-diphenyltetrazolium bromide) (Carl Roth) was added and the samples were allowed to form formazan crystals for 4 h at 37 °C. Subsequently, the cells were lysed with a solution containing isopropanol and hydrochloric acid (both from Carl Roth) and the absorbance was determined at 540–690 nm with a TECAN plate reader (Tecan, Männedorf, Switzerland). 

### 2.7. Soft Agar Clonogenic Assay 

GBM cells (500 cells/well for H4 and 2000 cells/well for U343) were mixed 1:1 *v*/*v* with 0.6% low-gelling agarose and were added over a 1% high-gelling agarose layer in 96-well plates. Both types of agaroses were from Carl Roth. The plates were incubated for 1 h at 4° C to allow the solidification of the low-gelling agarose. Subsequently, fresh culture medium was added in each well. The samples were cultured at 37 °C for 10 days (H4 cells) or 11 days (U343 cells) with medium change every 3–4 days. To better visualize the colonies, the samples were stained with 0.05% Crystal Violet in PBS (both from Carl Roth) for 1 h at room temperature. The colonies were counted with a BZ-X810 microscope (Keyence, Neu-Isenburg, Germany). Only colonies with a diameter of at least 50 µm were included in the analysis. 

### 2.8. Invasion Assay

To assess GBM invasion, we used the ORIS^TM^ system (Platypus Technologies LLC, Madison, WI, USA) according to the manufacturer’s protocol. Briefly, 96-wells were coated with a matrix containing 1 mg/mL rat tail collagen I and a ‘gap’ was created in the centre of each well using silicone stoppers. GBM cells were added to each well and allowed to adhere overnight. After removal of the stoppers, a second layer of 1 mg/mL collagen I was added so that the cells were completely embedded in this matrix. Brightfield micrographs of the ‘gap’ were taken at 0 days (pre-invasion status), 2 days (post-invasion for H4 cells), and 6 days (post-invasion for U343 cells). The area of the cell-free zone was measured with the Image J 1.48v software. The degree of ‘gap’-closure was calculated according to the formula:1 ÷ (area pre-invasion × 100 ÷ area post-invasion) × 100

### 2.9. Cell Death/Survival Assay

GBM cells were seeded in 24-well plates at a density of 50,000 cells/well. The next day, the cells were exposed to the indicated doses of temozolomide (TMZ; Selleck Chemicals, Houston, TX, USA) for 72 h or erastin (Sigma-Aldrich, St. Louis, MO, USA) for 48 h. Neutrophils (10^6^ cells/mL) were stimulated with GBM supernatants for 24 h. All cells were stained using the FITC-Annexin V/PI detection kit (BioLegend, San Diego, CA, USA) according to the manufacturer’s recommendations. All samples were analysed by flow cytometry using a BD FACSCanto™ II cytometer (BD Biosciences, Heidelberg, Germany).

### 2.10. Chemotaxis Assay

Neutrophil chemotaxis was assessed using transwell inserts with 3 µm pores (Sarstedt, Nümbrecht, Germany). Briefly, 24-wells were filled with 800 µL GBM supernatants or medium control and the inserts were placed into the wells. Neutrophils (5 × 10^5^ cells/200 µL medium) were added to each insert. The number of migrated cells was determined after a 3 h incubation period at 37 °C. 

### 2.11. MMP9 Release

Neutrophils (10^6^ cells/mL) were stimulated with GBM supernatants or with medium control for 1 h at 37 °C. The release of MMP9 by neutrophils was analysed by gelatine zymography as previously described [20]. Quantification of the gelatinolytic bands was performed with the ImageJ 1.48v software.

### 2.12. Interleukin-8 Release

The levels of Interleukin-8 in the GBM supernatants were assessed with the Human IL-8/CXCL8 DuoSet ELISA kit according to the protocol provided by the manufacturer (R&D Systems, Minneapolis, MN, USA). Absorbance was assessed at 450–540 nm with a TECAN plate reader. The analysis was performed independently for different batches of supernatants.

### 2.13. Statistical Analysis

All data involving GBM patients were analysed with the SPSS statistical software version 28 (IBM Corporation, Armonk, NY, USA). Survival curves were generated using the Kaplan–Meier method and the initial significance was assessed through univariate analysis with the log-rank test. Subsequently, multivariate analysis using Cox’s proportional hazard models was conducted to ascertain the prognostic value of selected variables. Spearman’s rank test (Spearman’s Rho) was used for correlation analysis and the data were presented as scatter-plots. The difference regarding marker expression between the various groups of GBM patients was analysed using Box-Whisker plots and Mann–Whitney U-test. The data from the in vitro studies were statistically analysed using the paired Student’s *t*-test. The level of significance was set at *p* ≤ 0.05 in all studies. 

## 3. Results

### 3.1. PGRMC1 Expression and Clinical Relevance in Glioblastoma (GBM) Patients

In the first set of studies, we tested whether PGRMC1 associated with the overall survival and progression-free survival of IDH wild-type GBM patients (Hannover cohort: *n* = 135; Magdeburg cohort: *n* = 170). To this end, the expression level of PGRMC1 was dichotomized into ‘low’ and ‘high’ based on the median-split method. The survival curves were generated using the Kaplan–Meier method and statistical significance was initially determined with the log-rank test. In the Hannover cohort, GBM patients with high tumour levels of PGRMC1 had a significantly poorer overall survival compared to patients with low levels of PGRMC1 (*p* = 0.010; log-rank) (Figure 1A). These findings were confirmed in the Magdeburg cohort of GBM patients (*p* = 0.005; log-rank) (Figure 1B). In both cohorts, PGRMC1^high^ patients had a shorter progression-free survival compared to their PGRMC1^low^ counterparts; however, statistical significance was only reached in the Magdeburg cohort (*p* = 0.283 and *p* = 0.012, respectively; log-rank) (Figure 1C,D). 

The overall survival of GBM patients was further analysed using Cox proportional-hazard models adjusted for potential confounders, such as age, Karnofsky Performance Scale (KPS), extent of surgical resection, and therapy [24,25,26,27]. In both cohorts, high expression of PGRMC1 significantly predicted the shorter overall survival of GBM patients (Hannover cohort: HR = 1.532, CI [95%] = 1.042–2.253, *p* = 0.030; Magdeburg cohort: HR = 1.462, CI [95%] = 1.039–2.057, *p* = 0.029) (Figure 1E). These data indicate that PGRMC1 could serve as an independent prognostic biomarker for the overall survival of patients with this type of cancer. Whether PGRMC1 may be a suitable biomarker for the progression-free survival of GBM patients requires, however, further clarification on additional patient cohorts.

### 3.2. PGRMC1 and GBM Tumour Cell Functions

To assess the role of PGRMC1 in GBM biology and functions, we used two different GBM cell lines stably transfected to downregulate PGRMC1 (sh-PGRMC1; see Material and Methods section). We first determined the metabolic activity of transfected GBM cells by MTT assay. The results showed that downregulation of PGRMC1 significantly decreased the metabolic activity in both cell lines starting at three days (H4 cells) and four days (U343 cells) in culture (Figure 2A,B). We additionally determined the anchorage-independent growth of transfected GBM cells by allowing the cells to form colonies in low-gelling agarose for 10 days (H4 cells) and 11 days (U343 cells). In both models, sh-PGRMC1 cells formed significantly less colonies compared to their control-transfected counterparts (Figure 2C). A representative example of colony formation by the H4 cells is shown in Figure 2D.

Furthermore, we tested whether PGRMC1 regulates the invasiveness of GBM cells. The invasion of sh-PGRMC1 versus sh-control cells was assessed by the degree of ‘gap’-closure in a 3D collagen matrix using the Oris^TM^ system. The results showed that PGRMC1 knockdown significantly decreased the invasiveness of both H4 and U343 cells (Figure 2E). A representative example of pre-invasion (day 0) versus post-invasion (day 2) status of the H4 cells is shown in Figure 2F. Together, these data indicate that PGRMC1 promotes the malignancy of GBM cells by enhancing their proliferation, anchorage-independent growth, and invasiveness.

### 3.3. Molecular Mechanisms of PGRMC1 in GBM

Next, we sought to characterize which molecular mechanisms are associated with PGRMC1 in GBM cells. To this end, we assessed by Western blot the protein expression of several markers known to regulate tumour proliferation and invasion, such as Cyclin D3, CDK 2/4/6, TCF 1/7, Integrin beta-1, MT1-MMP, SNAIL, Vimentin, ZEB1, and ZO-1. The results showed that the levels of Integrin beta-1 (ITGB1) were significantly lower upon PGRMC1 knockdown in both H4 and U343 cell lines (Figure 3A,B). Similarly, sh-PGRMC1 cells had significantly less TCF 1/7 compared to their control-transfected counterparts (Figure 3C,D), though it should be mentioned that the baseline expression of TCF 1/7 was relatively weak in both cell lines.

To determine the clinical relevance of these findings, we stained ITGB1 and TCF 1/7 in tissues from 165 GBM patients. ITGB1 significantly correlated with PGRMC1 in these tissues (*p* = 0.002, Rho = 0.244; Spearman’s rank) (Figure 4A) and PGRMC1^high^ GBM patients had significantly higher levels of ITGB1 compared to PGRMC1^low^ patients (*p* = 0.005, Mann–Whitney U) (Figure 4B). Representative examples of GBM tissues with synchronous low and high levels of PGRMC1/ITGB1 are shown in Figure 4C,D, respectively and at higher magnification in Appendix A. TCF 1/7 was only weakly and scarcely expressed in GBM tissues (less than 10% positive samples), which suggests that its relevance in the pathophysiology of GBM would be rather low, at least in the context of PGRMC1 signalling.

### 3.4. PGRMC1 and GBM Therapy

Since PGRMC1 is known to promote chemotherapy resistance in different types of cancer [12,13,14,15,16], we sought to determine its role in the response to therapy of GBM. Transfected H4 and U343 cells were exposed to temozolomide (TMZ)––the standard chemotherapeutic agent for GBM––or to DMSO as control. The percentage of surviving cells was assessed 72 h later by Annexin-Propidium Iodide (PI) staining and flow cytometry. After initial dose titration studies in wild-type cells (Appendix A), we selected the dose of 500 µM TMZ for both cell lines. In both models, sh-PGRMC1 cells were more susceptible to TMZ treatment than their control-transfected counterparts, although statistical significance was only reached in the H4 cells (Figure 5A,B).

Interestingly, very recent studies showed that PGRMC1 promoted an iron-dependent type of cell death (ferroptosis) in head and neck cancer cells [18]. We, therefore, tested the response of PGRMC1-transfected GBM cells to the ferroptosis inducer erastin. Following an initial dose titration (Appendix A), the cells were exposed to 50 µM (H4) or 5 µM (U343) erastin and to DMSO as control. The percentage of surviving cells was assessed 48 h later as described above. The results showed that PGRMC1 knockdown cells survived significantly better upon erastin treatment compared to their control-transfected counterparts (Figure 5C,D). These findings indicate that high levels of PGRMC1 may render GBM cells less susceptible to the current standard chemotherapy with TMZ, but more susceptible to ferroptosis inducers.

### 3.5. PGRMC1 and the Immune Microenvironment in GBM

Recent evidence from our own and other groups indicates that GBM cells recruit neutrophils to the tumour site and stimulate them to acquire a pro-tumour phenotype [20,21,28,29]. To determine whether PGRMC1 plays a role in the interaction between GBM cells and neutrophils, we produced conditioned supernatants (SN) from sh-control and sh-PGRMC1 GBM cells. We subsequently determined neutrophil chemotaxis by allowing peripheral blood neutrophils to migrate towards these supernatants in a transwell system (Appendix A). As negative control, the neutrophils were allowed to migrate towards regular cell culture medium. Additionally, we stimulated neutrophils with the GBM supernatants and assessed their survival as well as the release of MMP9 (Appendix A). Neutrophils incubated in medium only served as negative control. The results showed that supernatants from the H4 sh-control cells induced neutrophil chemotaxis at 3 h post-incubation. This effect was significantly lower in the presence of H4 sh-PGRMC1 supernatants (Figure 6A). We sought to confirm these findings with the U343 cell line; however, these cells did not have a stimulatory effect on neutrophils. We, therefore, used another GBM cell line for validation (U251), which we have previously found to modulate neutrophil biology [20]. Similar to H4, supernatants from the U251 sh-control cells induced neutrophil chemotaxis, while those from the U251 sh-PGRMC1 cells had a significantly weaker chemotactic potential (Figure 6A). The supernatants from both GBM cell lines also enhanced neutrophil survival and the release of MMP9; however, PGRMC1 knockdown had only a minor effect on these biological functions of neutrophils (Figure 6B,C). Taken together, these data suggest that PGRMC1 mainly modulates the production of chemoattractants by the GBM cells. To test this hypothesis, we determined the levels of Interleukin-8 (CXCL8)—a key neutrophil chemokine—in the GBM supernatants. The results showed that both GBM cell lines produced high levels of Interleukin-8, which were significantly reduced upon PGRMC1 knockdown (Figure 6D).

Based on these findings, we postulated that GBM tissues with a high PGRMC1 expression would recruit more neutrophils than tissues with low levels of PGRMC1. We, therefore, determined neutrophil infiltration in our GBM tissues by staining for the neutrophil marker CD66b. Tissues with at least five CD66-positive cells per microscopic field at 20× magnification were considered as CD66b^pos^; the rest as CD66b^neg^. Representative micrographs of CD66b^pos^ and CD66b^neg^ GBM tissues are shown in Figure 6E. We found a direct and significant correlation between PGRMC1 expression and CD66b positivity in all GBM patients (*p* = 0.002, Rho = 0.182; Spearman’s rank). This correlation was even stronger when only tissues with very low (H-score ≤ 150) and very high (H-score ≥ 250) PGRMC1 expression were included in the analysis (*p* < 0.001, Rho = 0.363; Spearman’s rank). A graphical representation of the latter shows clearly that the majority of patients with PGRMC1 ≤ 150 are also CD66b^neg^ while the majority of patients with PGRMC1 ≥ 250 are CD66b^pos^ (Figure 6F).

## 4. Discussion

A large number of studies sought to characterize the cellular and molecular factors that modulate the pathophysiology of glioblastoma (GBM), but the search continues. Our study identifies PGRMC1 as a tumour-promoting factor and modulator of the tumour microenvironment that may serve as an independent prognostic marker for the overall survival of GBM patients and, most importantly, as a target for novel therapeutic strategies in GBM.

Accumulating evidence links PGRMC1 to carcinogenesis and tumour progression. PGRMC1 is overexpressed in tumour tissues compared to non-malignant control tissues in multiple solid cancers (reviewed in [6]). Recent studies found an association between high gene expression of *PGRMC1* and poor survival of patients with hepatocellular carcinoma [8], triple-negative breast carcinoma [9], and head-neck squamous cell carcinoma [11]. Furthermore, high protein levels of PGRMC1 significantly associate with the shorter survival of renal carcinoma patients [10]. Our studies on two independent GBM cohorts showed that patients with high levels of PGRMC1 (PGRMC1^high^) had a significantly shorter overall survival compared to PGRMC1^low^ patients. These findings prompted us to characterize the role of PGRMC1 in the biology and functions of GBM cells.

Our data on two different PGRMC1 knockdown in vitro models demonstrate that PGRMC1 promotes the metabolic activity and the anchorage-independent growth of GBM cells. These findings are supported by previous studies showing that PGRMC1 modulates the proliferation of breast and oral carcinoma cells. Specifically, Pedroza and colleagues showed that PGRMC1 silencing decreased proliferation and induced cell cycle arrest in ER-positive cancer cells, while PGRMC1 overexpression resulted in increased proliferation of non-tumourigenic MCF10A cells [30]. Similarly, Asperger and co-workers found that downregulation of endogenous PGRMC1 decreased the viability of MCF7 and T47D cells, while PGRMC1 overexpression led to formation of larger tumours in murine xenograft models [31]. Recent studies from Huang et al. additionally showed that PGRMC1 was essential for the proliferation of oral cancer cells by inducing cell-cycle entry into the G2/M phase [32]. Our study further demonstrates that PGRMC1 promotes the invasion of GBM cells. A link between PGRMC1 and tumour migration/invasion has been observed in other types of cancer as well [30,32], but this phenomenon remains poorly understood thus far. Nevertheless, these findings are of particular importance for the pathophysiology of GBM, since the high invasiveness of these tumours and their subsequent infiltration into the adjacent brain tissue is the main cause for the low success rate of therapeutic surgery.

Previous molecular studies found a clear association between PGRMC1 and the EGFR signalling pathway in cancer. Specifically, PGRMC1 dimerizes in a heme-dependent manner and subsequently binds to EGFR, thereby enhancing tumour aggressiveness [17]. Furthermore, *PGRMC1* correlates with EGFR at mRNA level in patients with hepatocellular carcinoma [8]. PGRMC1 was also reported to promote progression of breast cancer via HER2 clustering in lipid rafts [31]. Using proteomic and RNA interference approaches in oral cancer cell models, Huang and co-workers showed that PGRMC1 induced epithelial-mesenchymal transition (EMT) via SIP1, SNAIL, and Twist in these cells [32]. Our initial screening for potential downstream targets of PGRMC1 in GBM did not reveal an association with the EMT markers SNAIL, Vimentin, ZEB1, and ZO-1. However, these findings do not exclude that PGRMC1 modulates the EMT process in GBM since (1) there are additional EMT markers, which we have not yet tested, and (2) EMT requires paracrine/autocrine inducers that may not be present in our cell models. We did find an association of PGRMC1 with Integrin beta-1 (ITGB1) and TCF 1/7, as these proteins were significantly downregulated upon PGRMC1 knockdown in GBM cells. Notably, the expression levels of PGRMC1 and ITGB1 also correlated significantly in tissues from GBM patients, which underlines the relevance of the PGRMC1/ITGB1 axis for the pathophysiology of GBM in vivo. Furthermore, these data support our functional studies, since beta-integrins are critical modulators of tumour proliferation and invasion in many types of cancer including gliomas (reviewed in [33,34]).

Perhaps the best-characterized role of PGRMC1 in cancer is modulation of resistance to chemotherapy. Already 17 years ago, Crudden and co-workers showed that PGRMC1 protected breast cancer cells from doxorubicin and camptothecin-induced cytotoxicity [35]. Since then, a number of studies demonstrated that PGRMC1 induced drug resistance in a variety of cancers, including breast, ovarian, uterine, colon, and lung cancer [12,13,14,15,16,17]. In line with these studies, our findings suggest that PGRMC1 renders GBM cells less susceptible to the alkylating agent temozolomide (TMZ), which is currently the gold standard chemotherapeutic drug in GBM. However, since the lower proliferation rate of the PGRMC1 knockdown cells may also affect their response to TMZ treatment, these results need to be validated by more comprehensive functional and molecular studies. Interestingly, very recent studies from You and colleagues found that PGRMC1 expression increased sensitivity of head and neck cancer cells to ferroptosis inducers both in vitro and in vivo [18]. This effect was observed in our own studies, where PGRMC1 knockdown GBM cells were significantly more resistant to erastin-induced cell death compared to control transfected cells. While erastin itself has only been used in pre-clinical models so far, there are several FDA/EMA-approved drugs, like sulfasalazine and sorafenib, which can induce ferroptosis (reviewed in [36]). Taken together, these data indicate that PGRMC1 may be a therapeutic target in GBM and that patients with high tumour expression of PGRMC1 might benefit from individualized therapeutic approaches with ferroptosis inducers.

Accumulating evidence indicates that the immune microenvironment is a key modulator of tumour biology and determines cancer progression. The GBM microenvironment is often infiltrated by neutrophil granulocytes, which display a pro-inflammatory and tumour-promoting phenotype [20,21,28,29]. Thus, it is not surprising that high neutrophil infiltration associates with a poor outcome in GBM patients [19,20,21]. In order to infiltrate the tumour tissue, neutrophils first need to be recruited from the peripheral blood. Lee and colleagues showed that neutrophil recruitment could be induced via a CD133-Interleukin 1-beta signalling axis in GBM [28]. Zha and co-workers identified a positive feedback loop in which neutrophil extracellular traps (NETs) activated NF-kB in glioma cells leading to Interleukin-8 secretion, with subsequent neutrophil recruitment and NETs formation [21]. Our study indicates that PGRMC1 also plays a role in neutrophil recruitment, likely via modulation of Interleukin-8 production by GBM cells. Interestingly, recent studies in hepatocellular carcinoma showed that PGRMC1 activated the NF-kB pathway to promote the release of Interleukin-6 [8]. While this phenomenon still requires characterization, it would be tempting to speculate that a similar mechanism mediates PGRMC1-induced Interleukin-8 release in GBM.

## 5. Conclusions

Our study identifies PGRMC1 as an independent prognostic biomarker in GBM. We further demonstrate that PGRMC1 mediates critical biological functions of GBM cells such as proliferation and invasion, possibly via ITGB1. Importantly, we link PGRMC1 with GBM cell resistance to TMZ-induced cytotoxicity but with increased susceptibility to ferroptosis. Finally, we provide first evidence that PGRMC1 can modulate the immune microenvironment in GBM. These findings contribute to a better understanding of GBM pathophysiology and may ultimately foster the development of novel therapeutic strategies against this type of cancer.

## Figures and Tables

**Figure 1 cells-12-02498-f001:**
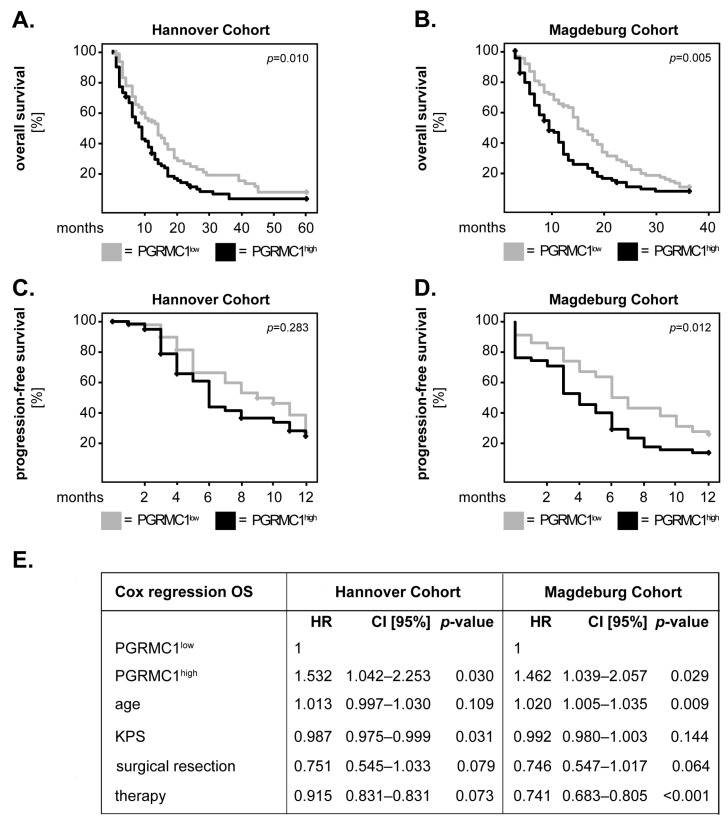
PGRMC1 in GBM patients: univariate and multivariate survival analysis. PGRMC1 expression was dichotomized into ‘low’ and ‘high’ according to the median-split method. Kaplan–Meier curves were plotted for the (**A**) five-year overall survival in the Hannover cohort, (**B**) three-year overall survival in the Magdeburg cohort, and (**C**,**D**) one-year progression-free survival in both cohorts. The log-rank test was used for statistical analysis and the *p*-values are indicated in the upper-right corner of each plot. (**E**) Multivariate Cox regression analysis model for the overall survival of patients with high versus low levels of PGRMC1. HR: hazard ratio; CI [95%]: 95% confidence interval.

**Figure 2 cells-12-02498-f002:**
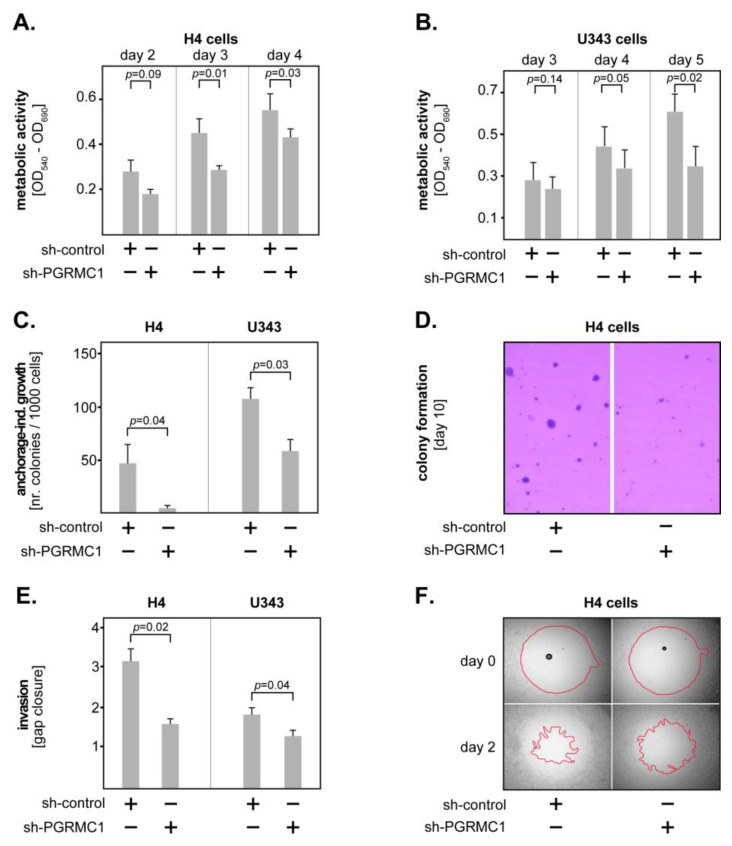
PGRMC1 modulates GBM proliferation, adhesion-independent growth, and invasion. PGRMC1 knockdown significantly inhibited the metabolic activity of (**A**) H4 and (**B**) U343 GBM cells, as indicated by the MTT assay. (**C**) PGRMC1 knockdown significantly inhibited the anchorage-independent growth of H4 and U343 cells. (**D**) Representative micrographs of colonies generated by the H4 cells after 10 days of culture in low-gelling agarose. (**E**) PGRMC1 knockdown significantly inhibited the invasiveness of H4 and U343 cells. (**F**) Representative micrographs of an invasion assay of the H4 cells showing the pre-invasion status (upper panels) and the post-invasion status (lower panels). The red lines mark the closure of the ‘gap’, indicating the degree of tumour invasion. At least three independent experiments were performed for each assay. Shown are the means + S.D. In all studies, statistical analysis was performed with the paired *t*-test.

**Figure 3 cells-12-02498-f003:**
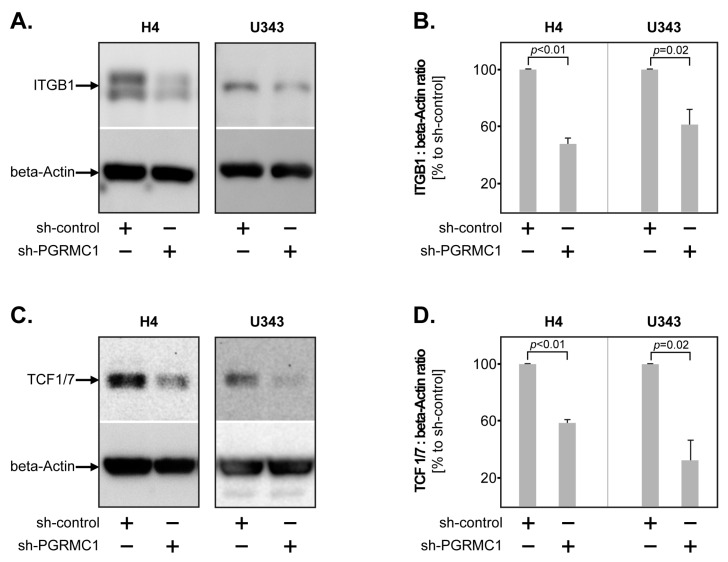
ITGB1 and TCF 1/7 are downstream of PGRMC1 in GBM. (**A**) Representative Western blot of ITGB1 levels in sh-PGRMC1 versus sh-control GBM cells. Beta-Actin was used as loading control. (**B**) PGRMC1 knockdown significantly decreased the levels of ITGB1 in both H4 and U343 cells. (**C**) Representative Western blot of TCF 1/7 levels in sh-PGRMC1 versus sh-control GBM cells. Beta-Actin was used as loading control. (**D**) PGRMC1 knockdown significantly decreased the levels of TCF 1/7 in both H4 and U343 cells. Shown are the means + S.D. of at least three independent experiments. Statistical analysis was performed with the paired *t*-test.

**Figure 4 cells-12-02498-f004:**
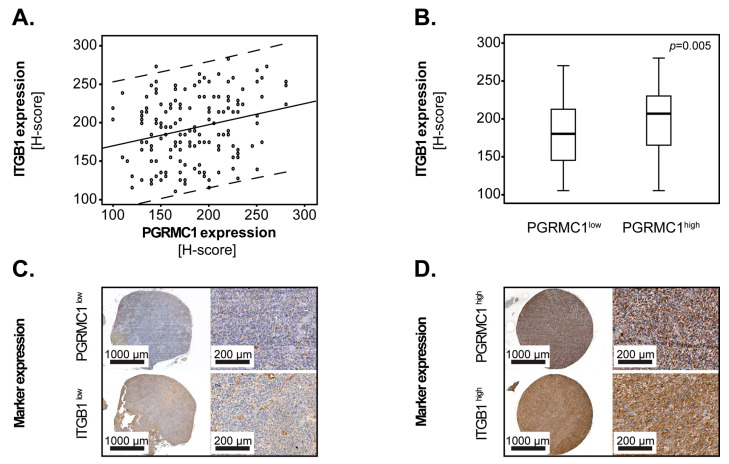
The levels of PGRMC1 and ITGB1 correlate in GBM patients. (**A**) Scatter plot analysis of PGRMC1 and ITGB1 expression levels in tumour tissues from GBM patients. Dotted lines indicate the 95% confidence interval for the regression line. Statistical analysis was performed with Spearman’s rank test. (**B**) ITGB1 expression in patients with low versus high levels of PGRMC1. Shown are the medians (black lines) and percentiles (25th and 75th) as vertical boxes with error bars. Statistical analysis was performed with the Mann–Whitney U test and the *p*-value is indicated in the upper-right corner of the plot. Representative micrographs showing (**C**) synchronous low levels and (**D**) synchronous high levels of PGRMC1 and ITGB1 in GBM tissues. The scale bars are indicated in the lower-left corner of each panel.

**Figure 5 cells-12-02498-f005:**
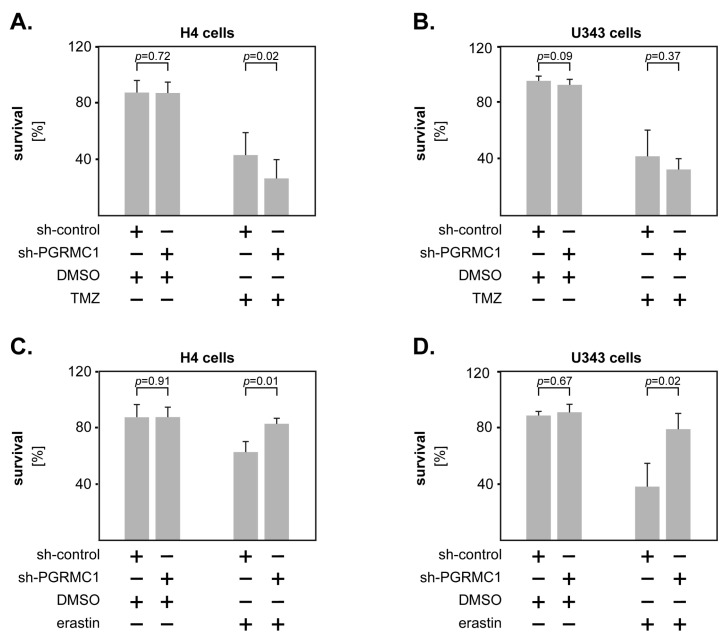
PGRMC1 modulates the response to therapy of GBM cells. (**A**) H4 and (**B**) U343 cells were stimulated with 500 µM temozolomide (TMZ) or DMSO for 72 h. The percentage of surviving cells (Annexin V-negative/PI-negative) was assessed by flow cytometry. GBM cells were stimulated with (**C**) 50 µM erastin—for H4 and (**D**) 5 µM erastin—for U343. DMSO was used as control and the percentage of surviving cells was assessed 48 h later by flow cytometry. Shown are the means + S.D. of at least three independent experiments. Statistical analysis was performed with the paired *t*-test.

**Figure 6 cells-12-02498-f006:**
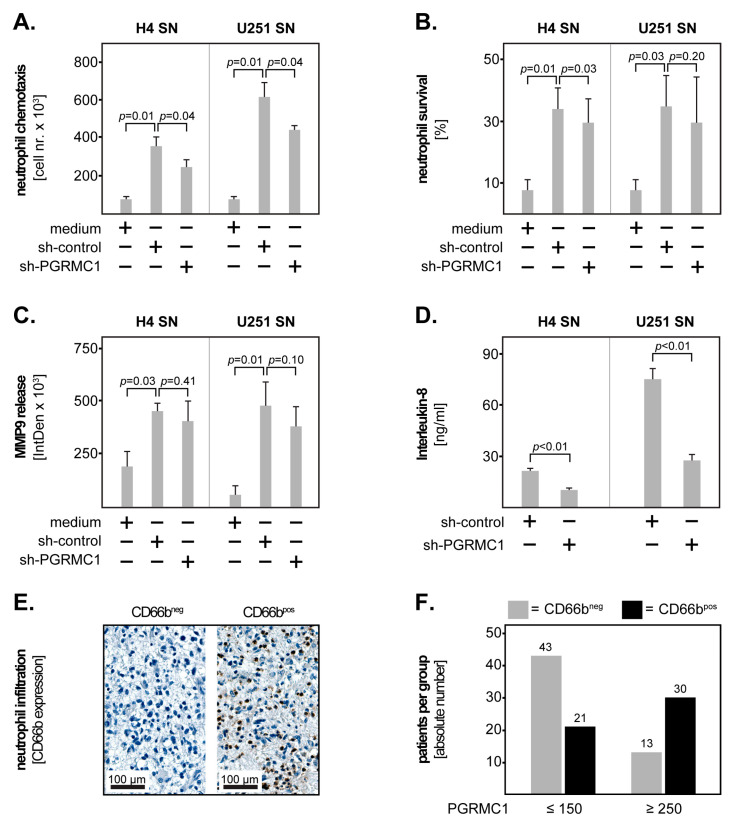
PGRMC1 modulates the GBM-neutrophil interactions. Effect of conditioned supernatants (SN) from sh-control and sh-PGRMC1 GBM cells on (**A**) neutrophil chemotaxis, (**B**) neutrophil survival, and (**C**) the release of MMP9 by neutrophils. Regular cell culture medium was used as control. Neutrophils from three (**A**,**C**) and four (**B**) independent blood donors were included in the respective assays. (**D**) Interleukin-8 levels in SN from sh-control and sh-PGRMC1 GBM cells as determined by ELISA. Shown are the means + S.D. of three independent SN batches. Statistical analysis was performed with the paired *t*-test. (**E**) Representative micrographs of GBM tissues with (CD66^pos^) or without (CD66b^neg^) tumour-infiltrating neutrophils. The scale bars are indicated in the lower-left corner of the panels. (**F**) Numbers of CD66b^neg^ (grey bars) and CD66b^pos^ (black bars) GBM patients in the groups with very low (H-score ≤ 150) versus very high (H-score ≥ 250) PGRMC1 expression.

## Data Availability

The datasets for the in vitro studies are included within the article and its Appendix A. The datasets on patient material are not publicly available as they also contain individual participants’ details. These data can be obtained from the corresponding author in anonymized form upon reasonable request.

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
