# Peer review of "Progesterone Receptor Membrane Component 1 (PGRMC1) Modulates Tumour Progression, the Immune Microenvironment and the Response to Therapy in Glioblastoma"

_cells, 2023, doi:10.3390/cells12202498_

Round 1

Reviewer 1 Report

The authors explored the expression and associations of Progesterone Receptor Membrane Component 1 (PGRMC1) with various features of glioblastoma (GBM) to uncover its role in pathophysiology. They used two cohorts of IDH wild-type GBM patients (Hannover cohort: n=135; Magdeburg cohort: n=170) and GBM cells with stable PGRMC1 knockdown. The rationale for studying the gene is weak, I suspect the gene has been selected from earlier studies of genes overexpressed in GBM in TCGA cohorts. Any gene overexpressed in a WHO grade G4 would show associations with various pathological features. They show that PGRMC1 promotes the proliferation, anchorage-independent growth and migration of GBM cells. Integrin beta-1 (ITGB1) and TCF 1/7 were reported as downstream of PGRMC1 based on their lower levels in PGRMC1 depleted cells. High expression of PGRMC1 rendered GBM cells less susceptible to temozolomide (TMZ) in one cell line, but more susceptible to the ferroptosis inducer erastin. They studied neutrophil chemotaxis under influence of conditioned supernatants (SN) from sh-control and sh-PGRMC1 GBM cells. They report PGRMC1 enhanced Interleukin-8 production in GBM cells and promoted the recruitment of neutrophils. The expression PGRMC1 significantly correlated with the numbers of tumor-infiltrating neutrophils also in tissues from GBM patients.

Standard computational methods have been applied and the results have a limited novelty and scope. While some of the data are of interest and could be a starting point for mechanistic studies, most what is presented are associations and very superficial analysis. Technical details of some functional experiments are insufficient. There are also experimental concerns that call this work into question.

The interpretation of the results goes too far considering a limited scope of data validation. Some biochemical studies were performed with PGRMC1 depleted cells but the tests were not optimal. It takes more studies to make a gene overexpressed in GBM a biomarker.

The language of the manuscript is acceptable but the large extent comprising long stretches of the text show unacceptable 48% similarity to previous authors’ study and other publications. This refers not only to methods that could have strong similarity, but also to results and discussion. It looks like a name of the gene of interest was changed and the most of the text comes from the previous study which is an unacceptable situation.

Main comments:

1)     If the PGRMC1 is upregulated in gliomas, all detected associations would reflect an association of a high grade with tested parameters. The analysis is therefore not informative as it is known what are GBM associations. However, statistical significance was only reached in the Magdeburg cohort., so The authors should check PGRMC1 expression in other large cohorts (TCGA, GCCA, REMBRANT). Please write a gene name in italics to distinguish from the protein.

2)     Confirmation of stable knockdown of PGRMC1 in GBM cell lines should be presented in main figures , not in supplementary (fig. S2) with all controls, number of replicates and statistics for mRNA and protein levels.

3)     The applied test is rather a wound closure test determining cell migration. Invasion is tested if the cells passed the matrigel and insert in a chamber. Was a serum present? Glioma cell in vitro are low invasive if serum or microglia are not added to the system. In fig.2E Y axis is described as invasion (gap closure). What is a unit, what it represents? How this was calculated?

4)     If the gene knockdown reduces proliferation and colony growth, the degree of ‘gap’- closure in a 3D collagen matrix is not appropriate to study tumor invasion as the test is sensitive to a cell number and cell proliferation, particularly when cell behavior is studied for longer times: 18 or 48 hr. However, if the knockdown affect cell proliferation it would delay cell migration and invasion as both tests dependent on counting cell number. To prove that knockdown of PGRMC1 affects invasion more evidence is required, along with some mechanistic data (i.e., MMPs activities by zymography). The conclusion should be moderated.

5)     Fig.3A. Immunoblots look poor, they are too processed (original blots look much better) and the white box in the middle should be removed. The statement that ITGB1 and TCF 1/7 are downstream of PGRMC1 in GBM is misleading, their levels are lower in KD cells but they do not to be downstream of The expression and impacts of ITGB1 in GBMs are well known.

6)     Technical details are too vague to reproduce the study: i.e., qPCR Gene expression analysis do not state PCR thermal conditions, quantities of reagents. The information about normalization and way of calculation the expression is not sufficient. It is not clear if any controls for immunohistochemistry on TMAs were applied.

7)     Figure 4. It is not clear how many samples and from cohort or TMAs the data originate.

8)     Figure 5. The effects of PGRMC1 knockdown on sensitivity to TMZ are visible only in one cell lines and despite large variations. Besides due to slower proliferation of KD cells the response would be different as there is less cells after 72 h. Knockdown sensitized both cell lines to erastin. The wording “cells were stimulated with temozolomide (TMZ)” is not correct as cells are treated or exposed to TMZ. The process is not a stimulation involving some kind of receptors.

9)     They report that peripheral blood neutrophils migrate towards these supernatants in a transwell system. Were those cells PMBC because there is no information about separating neutrophils? The fig. 5 shows different neutrophil chemotaxis in the presence of KD media in comparison to sh-cnt cells. I wonder at the statistical difference that is claimed regarding large variations. They state that cells from more than 3 donors were used but the precise number should be provided. The authors used the paired t-test but with 3 conditions they should use Anova. Meaning of the panel E is not clear and data in the panel F shows quantification but without any statistical analysis and numbers of samples in quantification.

Reviewer 2 Report

The manuscript describes elegantly the investigation of a possible role for PGRMC1 in the biology of GBM. The results are well presented and analyzed, pointing to a possibly relevant role for the molecule, both as a prognostic marker and a potential target for new therapeutic approaches. The only suggestion I would make would be to enlarge the histopathological images in figure 4 C and D, which, at their current size, do not allow a precise characterization of cell staining.

Author Response

Comment: The manuscript describes elegantly the investigation of a possible role for PGRMC1 in the biology of GBM. The results are well presented and analyzed, pointing to a possibly relevant role for the molecule, both as a prognostic marker and a potential target for new therapeutic approaches. The only suggestion I would make would be to enlarge the histopathological images in figure 4 C and D, which, at their current size, do not allow a precise characterization of cell staining.

Reply: We thank the reviewer for the positive evaluation of our study. According to the reviewer’s suggestion, we now provide micrographs at higher magnification in the new supplementary Fig. S3 A-B.

Reviewer 3 Report

 The authors demonstrated that PGRMC1 increases tumor related inflammation and stimulate the progression of GBM. Meanwhile the manuscript also found that this function could through ITGB1. The manuscript does provide inside look of this pathway. The manuscript is good overall, there are some minor things need to be improved. 

1.     Is there a Erastin induction titration curve and cell density curve available?

2.     Is there better quality images of the WB available?

3.     Need some grammar check and spelling check. 

Need some grammar check and spelling check 
